# Dampened Slow Oscillation Connectivity Anticipates Amyloid Deposition in the PS2APP Mouse Model of Alzheimer’s Disease

**DOI:** 10.3390/cells9010054

**Published:** 2019-12-24

**Authors:** Alessandro Leparulo, Mufti Mahmud, Elena Scremin, Tullio Pozzan, Stefano Vassanelli, Cristina Fasolato

**Affiliations:** 1Department of Biomedical Sciences, University of Padua, Via U. Bassi 58/B, 35131 Padua, Italy; alessandro.leparulo@unipd.it (A.L.); muftimahmud@gmail.com (M.M.); elena.scremin@studenti.unipd.it (E.S.); tullio.pozzan@unipd.it (T.P.); 2Neuroscience Institute-Italian National Research Council (CNR), Via U. Bassi 58/B, 35131 Padua, Italy; 3Venetian Institute of Molecular Medicine (VIMM), Via G. Orus 2B, 35129 Padua, Italy; 4Padua Neuroscience Center (PNC), University of Padua, Via G. Orus 2B, 35129 Padua, Italy

**Keywords:** Alzheimer’s disease, B6.152H, PS2APP, amyloid precursor protein, presenilin-2, amyloid-β, slow oscillations, local field potentials, functional connectivity, phase-amplitude-coupling

## Abstract

To fight Alzheimer’s disease (AD), we should know when, where, and how brain network dysfunctions initiate. In AD mouse models, relevant information can be derived from brain electrical activity. With a multi-site linear probe, we recorded local field potentials simultaneously at the posterior-parietal cortex and hippocampus of wild-type and double transgenic AD mice, under anesthesia. We focused on PS2APP (B6.152H) mice carrying both presenilin-2 (PS2) and amyloid precursor protein (APP) mutations, at three and six months of age, before and after plaque deposition respectively. To highlight defects linked to either the PS2 or APP mutation, we included in the analysis age-matched PS2.30H and APP-Swedish mice, carrying each of the mutations individually. Our study also included *PSEN2*^−/−^ mice. At three months, only predeposition B6.152H mice show a reduction in the functional connectivity of slow oscillations (SO) and in the power ratio between SO and delta waves. At six months, plaque-seeding B6.152H mice undergo a worsening of the low/high frequency power imbalance and show a massive loss of cortico-hippocampal phase-amplitude coupling (PAC) between SO and higher frequencies, a feature shared with amyloid-free PS2.30H mice. We conclude that the PS2 mutation is sufficient to impair SO PAC and accelerate network dysfunctions in amyloid-accumulating mice.

## 1. Introduction

Understanding which brain dysfunctions occur at the cell and circuit levels before the onset of the neurodegenerative process is a matter of great urgency in the fight against AD [1]. Dynamic markers which are suitable for identifying the onset of the disease and its temporal evolution are required not only for human diagnostics, but also for studies based on AD mouse models. Although transgenic (tg) mouse lines based on human mutations linked to Familial AD (FAD) cannot mimic the full spectrum of the human disease and new models are under investigation [2], they are largely accepted as AD amyloidosis/gliosis research models, and are widely used to test treatments capable of modifying disease progression and rescue memory functions [3]. The use of these mouse models is also relevant when considering two complementary aspects recently emerged in the AD field. Firstly, Aβ dysregulation plays a key role in initiating AD [4], with brain vulnerability being promoted by Aβ accumulation through connectivity alterations rather than plaque proximity [5,6,7,8]. Secondly, new results suggest that some sporadic forms of AD are characterized by cerebral mosaicism of several APP variants, expressed at a higher than normal dosage, including also mutations observed in FAD cases [9].

Numerous studies have addressed functional and biochemical alterations of neuronal cells in FAD mice in association with memory impairment and plaque deposition. In particular, by using electrophysiological approaches, both in vitro and in vivo, previous studies pointed to an excitatory/inhibitory imbalance in the hippocampus leading to an altered ratio between hyperactive and silent neurons [10,11]. Both hyper-excitability and hyper-synchronicity characterize the early stages in AD mouse models [12,13,14,15,16,17], as well as in patients [18,19].

When considering hippocampal oscillations, attention was mainly focused on theta and gamma frequencies, given that disruption of phase-amplitude coupling (PAC) between these bands has been linked to cognitive impairments [20,21]. Yet, oscillations in the slow-wave range characterize brain rhythmicity during slow-wave sleep (SWS) and anesthesia, and are at the basis of memory consolidation and information transfer during sleep and unconsciousness [22]. Remarkably, it was recently demonstrated that functional connectivity in the slow-wave range, as measured by mesoscale calcium (Ca^2+^) signals, is severely reduced in neocortex, thalamus, and hippocampus of two AD mouse models [23]. In particular, slow-wave (0.1–3 Hz) coherence and propagation between cortical areas are impaired in plaque-accumulating AD mice, both under anesthesia and SWS. Alterations in the slow frequency (<3 Hz) bands have further been implicated in AD [23], and their manipulation restores the functionality of brain circuits and enhances memory consolidation in mice [23,24] and humans [25].

Nevertheless, in AD mice, we lack a systematic analysis of brain network activity at different cortical levels during amyloidosis progression. The aim of this work is twofold: to identify specific dysfunctions of the brain network that occur in a well-characterized AD mouse model, i.e., the B6.152H line, before and after plaque deposition, and to disentangle the specific contribution of mutations in PS2 and APP. To this end, we simultaneously recorded local field potential (LFP) signals in the posterior parietal cortex (PPC) and dorsal hippocampal formation (HPF) of anesthetized mice using a linear multi-site silicon probe. It should be noted that the PPC belongs to the Default Mode Network (DMN) which has a strong and functional connection with the HPF, one of the first regions affected in AD, both in humans and mice [5,6,8]. The spontaneous electrical activity of WT mice was thus compared to that of the double tg B6.152H mice, which express both the human APP-KM670/671NL (Swedish) and the PS2-N141I mutations, and show the first appearance of plaques and gliosis in the cortex and HPF at six months of age [26,27]. These mice have been studied by Positron Emission Tomography (PET) [28,29] and Magnetic Resonance Imaging (MRI) [30], as well as through pharmacological, electrophysiological, and Ca^2+^ imaging approaches [12,31,32]. To identify the differences related to either the initial accumulation of Aβ or the deposition of plaques, we used mouse cohorts of three and six months of age, corresponding, respectively, to either the absence or presence of plaques and gliosis in B6.152H mice [12,26]. To highlight brain network dysfunctions linked exclusively to amyloidosis, we also analyzed the tg lines PS2.30H and APPSwe, which carry the same mutations in PS2 or APP individually. The first line shows neither accumulation of Aβ nor gliosis, but shares with the B6.152H line the early defects of Ca^2+^ homoeostasis [12,32], while the APPSwe line shows accumulation of Aβ very late [27]. Furthermore, the *PSEN2^−^*^/*−*^ knockout (PS2KO) mouse line was also included in this study to distinguish between the gain and loss of PS2 function [33]. In fact, for PSs, both gain- and loss-of-function effects have been implicated in AD and, more generally, in neurodegeneration [34]. By this approach, we identified unique features of network dysfunctions that characterize the disease progression in B6.152H mice, as well as features shared with either PS2.30H (but not PS2KO) or APPSwe mice. In particular, the PS2-N141I is sufficient to observe dampened SO PAC between the deeper cortical layer and the hippocampus. Furthermore, in young B6.152H mice, the rapid accumulation of Aβ but not of α,β-APP carboxy-terminal fragment (α,β-CTFs) is likely responsible for power imbalance in the low frequency range and loss of SO cortico-hippocampal connectivity. Overall, these features highlight the role played by the PS2 mutation and Aβ accumulation on brain spontaneous activity during the disease progression.

## 2. Materials and Methods

### 2.1. Animals

The homozygous tg mouse lines B6.152H (PS2APP), PS2.30H, and BD.AD147.72H (APPSwe) were kindly donated by L. Ozmen (F. Hoffmann-La Roche Ltd., Basel, Switzerland) [26,27]. In these lines, *APP* and *PSEN2* transgenes are driven by mouse Thy-1.2 and prion protein promoters, respectively. The tg line *PSEN2^−^*^/−^ (PS2KO) [33] was obtained from the CNR-EMMA repository (Rome, Italy). The B6.152H and PS2KO were generated in a C57BL/6J background. The other lines were originally backcrossed to C57BL/6J mice for seven or more generations (>95% C57BL/6J background). As a control, we used C57BL/6J (WT) mice. All the animals were reared in a SPF animal facility, in 12/12-h light/dark cycles, with free access to food and water. For each genotype, mice were from three- or six-month-old cohorts, with either one- or two-week-tolerance, respectively. Only female mice were used because in B6.152H, the pathological hallmarks occur earlier with respect to males [26]. All experimental procedures were performed according to the European Committee guidelines (decree 2010/63/CEE) and the Animal Welfare Act (7 USC 2131), in compliance with the ARRIVE guidelines, and were approved by the Animal Care Committee of the University of Padua and the Italian Ministry of Health (authorization decree 522/2018-PR).

### 2.2. Animal Preparation and Surgery

Animal preparation for surgery and recordings was performed as described [12,35]. Mice were anesthetized by intraperitoneal injection of urethane (1.5 g/kg, Sigma–Aldrich, Milan, Italy) dissolved in 0.9% NaCl, and a mixture of xylazine/tiletamine-zolazepam (XTZ) (Rompun 1 mg/Kg plus Zoletil 10 mg/Kg) dissolved in phosphate buffer. Anesthesia induction was done by an initial dose of urethane followed, 30 min later, by a single dose of XTZ. The absence of reaction to noxious stimuli (e.g., tail, hind paw and ear pinches) ensured a constant level of anesthesia. Coupling XTZ with urethane anesthesia substantially reduced the mouse death rate from 50 to 20%. After shaving the fur over the head, mice were restrained in a stereotaxic frame and the skull was exposed. A hole was drilled on the skull over the left hemisphere at the site for insertion of the multi-site recording probe. To locate the DG for probe insertion, we used landmarks on the skull of each mouse, according to the procedure described by Huang et al. [36]. The left hemisphere of the PPC was selected, since amyloid plaque deposition in mice is stronger in this hemisphere upon spreading from the lateral entorhinal cortex (LEC) [8]. The cavity over the skull was filled with Krebs-Ringer solution and an external Ag-Cl reference electrode was dipped within. At the end of the electrophysiological experiment, the mouse was euthanized by excess of anesthesia and the brain was dissected. The left hemisphere, intended for histological investigations, was fixed in 4% paraformaldehyde (PFA) in Tris-buffer solution (TBS) in mM: NaCl 150, Tris 50, pH 7.4 at room temperature (RT). The cortex and hippocampus from the right hemisphere were snap-frozen in liquid nitrogen for biochemical assays. Body temperature was kept at 37 ± 0.5 °C all the time, using a servo-controlled heating pad (ATC1000-World Precision Instruments, Inc., Friedberg, Germany).

### 2.3. Signal Acquisition

Spontaneous LFP activity was acquired by a linear 32-electrode-silicon probe (ATLAS Neuro Probe: E32-100-S1-L6-NT; pointy tip feature; 100 µm spaced electrodes; mean impedance 0.28 MΩ in Krebs’ solution) connected by a 32-channel head stage (Intant, RHD2000) and an SPI cable to the acquisition system (Open Ephys, OEps Tech, Lisbon, Portugal). The probe was inserted into the PPC and lowered up to 2.3 mm, so that the first 24 channels of the probe were recording from the deepest layer o α,f dentate gyrus (DG) up to the cortical layers. The LFP signals were visualized, recorded, and digitalized at 10 kHz through the open Graphic User Interface software supplied with the Open Ephys acquisition system. Together with the LFP signals, other physiological signals were simultaneously recorded to determine the animal’s health conditions. Heartbeat was monitored through electrocardiogram (ECG) recording, 10X amplified and filtered between 1 and 100 Hz by means of a DAM 50 Amplifier (World Precision Instruments, Friedberg, Germany). ECG positive and negative derivations were subcutaneously inserted into the forelimbs. Respiration-induced movements of the chest wall were converted in voltage fluctuations by the piezoelectric properties of the temperature probe (IT-23, World Precision Instruments, Friedberg, Germany). The respiration signal was amplified 100X and band-passed between 0.1 and 100 Hz by means of a DP-301 amplifier (Warner Instruments, Crisel, Rome, Italy). Physiological signals were digitalized at 10 kHz by a PCI-6071E I/O card (−0.5 to 0.5 V input range) combined with a BNC-2090 terminal block (National Instruments, Rome, Italy) in differential mode and recorded through a custom-made LabView (National Instruments, Rome, Italy) script. Upon reaching a stable level of anesthesia (see below for quantitative evaluation of anesthesia level), we recorded the spontaneous brain activity for 30–40 min.

### 2.4. Data Processing and Analysis

Data analysis of electrophysiological signals was performed offline in Matlab (Mathworks Inc., Natick, MA, USA) using custom-written scripts. Firstly, the individual data files containing the acquired signals from different channels were converted from Open-Ephys format to Matlab file format. The analysis of the acquired raw data consisted of a preprocessing step, in which raw signals were cleaned through the application of a gaussian filter to remove 50 Hz noise and its harmonics. The signals from the first 24 channels were filtered through the built-in non-causal zero-phase distortion filtering function (filtfilt.m), which processes the data in both forward and reverse directions to avoid any phase distortion, with coefficients from the built-in Butterworth transfer function (butter.m). Baseline drift was removed from all the signals (i.e., LFP, ECG and respiration) using a median estimation method. Secondly, the signals were low-pass filtered (filter order: 5; cut-off frequency: 190 Hz for LFP, 25 Hz for ECG, and 10 Hz for respiration) and down-sampled (using the downsample.m function) to 500 Hz, 50 Hz and 20 Hz, respectively. Finally, in the LFP recordings, windows were selected for the analyses based on the stability of heart and respiration rates. Stable LFP windows were determined using the respiration and ECG signals in three steps. In the first step, signal parts showing anomalous respiration and heartbeat patterns were identified by calculating their rates and thresholding these rates using specific upper and lower bounds. The boundary values were calculated by taking the median of the individual rates and adding (for upper bound) and subtracting (for lower bound) heuristically-selected rate constants whose values were 0.5 and 2 for respiration and heartbeats, respectively. In the second stage, the stable signal parts were separated from the whole respiration and ECG signals, and overlapping signal portions were identified where both the signals were stable. In the final step, time-locked LFP signal windows corresponding to the overlapping stable signal portions were extracted and a five-min window was selected for all the analyses. This preprocessing step was necessary to ensure that all the animals were stable during the acquisition procedure. Histological confirmation of recording sites was sampled by DiI staining.

See Appendix A for electrophysiological data analyses, histology, and biochemistry.

### 2.5. Statistical Analyses

Data were expressed as mean ± SEM. Significance was evaluated by the non-parametric Kruskal-Wallis test with the Dunn-Sydak correction for multiple comparisons (k = 4, number of tg lines compared to age-matched WT mice). Where the Kruskal-Wallis test resulted in the existence of a pair of different populations, differences between means were tested with the Mann-Whitney Rank Sum test (* *p* < 0.05; ** *p* < 0.01). Boxplots show median, mean, the 25th and 75th percentiles of the distributions; whiskers indicate the upper and lower extremes. The analyses reported below were based on ten mice for each age cohort, with mice derived from at least three different litters. The sample size was chosen on the basis of pilot experiments and previously published data [12].

## 3. Results

### 3.1. In Vivo Multi-Site LFP Recordings from WT and Tg Mice

We focused our analysis on the double tg B6.152H mice that express the human APP-Swedish and PS2-N141I mutations. Parallel experiments were carried out in WT, as well as single tg PS2.30H and APPSwe mice [12,26,27], to disentangle the specific contributions of PS2 and APP mutations. The *PSEN2^−/−^* (PS2KO) mouse line [33] was also included to check for PS2 loss-of-function defects [34]. Of these lines, at six months of age, only the B6.152H shows plaque accumulation and gliosis in the cortex and hippocampus (Figure 1A–C), as previously reported [12], with no signs of neurodegeneration [26,27,29,33]. As shown in Figure 1D, from three to six months, there is 2-log increase in the Aβ42 level in the hippocampus of B6.152H mice, but only 1-log change in APPSwe mice, which also starts from a lower level compared to B6.152H mice. Of note, at three months, B6.152H mice have Aβ42 levels similar to those found in six-month-old APPSwe mice (around 100 pg/mg wet tissue), yet with considerably different Aβ42/Aβ40 ratios, being close to 1 in B6.152H and 0.2 in APPSwe mice, as previously reported [12,26]. Furthermore, in B6.152H mice, the ratios between hippocampal and cortical Aβ42 levels were significantly higher with respect to APPSwe mice (1.12 ± 0.03 and 0.92 ± 0.04, mean ± SEM, *n* = 4, *p* < 0.01), indicating that the most affected region is the hippocampus in young B6.152H mice, while it is the cortex in APPSwe mice.

To identify network dysfunctions linked to AD progression, we recorded spontaneous brain activity using a multi-site linear silicon probe with 24 microelectrodes sampling from the PPC to the HPF (Figure 2A) of mice anesthetized with a mixture of urethane and XTZ. Appendix A shows the general procedure used for probe insertion with the average coordinates, as well as the mean heart and respiration rates (Appendix A) of WT and B6.152H mice. No statistically significant difference was found between age-matched cohorts, suggesting a similar anesthesia level and skull anatomy.

For each recording site, we estimated the total power from the integral of the power spectral density (PSD) in the 0.1–190 Hz range (Appendix A). For the analyses, we selected frequency ranges on the basis of the power distribution in WT mice. In addition, because of the relevance of slow-waves in memory consolidation during SWS [37,38,39,40], the low wave range was subdivided in slow oscillations (SO) (0.1–1.7 Hz) and delta waves (1.7–4.7 Hz). The delta band corresponds to the theta band used with urethane anesthesia [12,41], and the frequency ranges here employed are those most commonly found in the AD studies [24,42] (Appendix A).

For a quantitative comparison of the power changes in the spontaneous electrical activity, we chose seven channels corresponding to peaks in the total power profile (Figure 2A,B), which were located at the following depths: (i) 2100 and 1900 μm, in the lower and upper *stratum granulare* (*lo-sg* and *up-sg*) of DG; (ii) 1500 and 1200 μm, in *stratum radiatum-lacunosum molecolare* (*sr-lm*) and *stratum pyramidale* (*sp*) of CA1; (iii) 900, 600, and 300 μm in Layer 6 (L6), L4/5 and L2/3 of the PPC. From the total power profile, we observed some power reduction in the HPF of three-month-old B6.152H, APPSwe and PS2.30H mice (Figure 2C). Only in the latter, the reduction reached statistical significance in DG, *sg* (−74%) and L4/5 (−50%) (*p* < 0.05, Figure 2C and Appendix A). The decrease of total power was spread across different frequency bands, especially affecting SO in DG (*p* < 0.01 *up-sg*, Appendix A). For both WT and tg mice, the total power profiles were stable while moving from three to six months of age, with no statistical differences at the selected sites (data not shown).

### 3.2. Power Imbalances Characterize B6.152H Mice

Comparisons in the different frequency bands were based on the relative band power, measured as the percentage of the mouse total power at each depth. Figure 3A reports the relative power in the different frequency ranges occurring in CA1 (*sr-lm*, 1500 μm) of three-month-old B6.152H, with respect to age-matched WT mice. The ring-chart shows a strong reduction in SO (red) and a concomitant increase in delta (yellow) relative powers. With respect to WT mice, the alterations in the relative power distribution represent power imbalances. In particular, as an index of power imbalance in the Low frequency range (0.1–4.7 Hz), we measured the power ratio between SO and delta band, which was significantly reduced only in B6.152H mice (−54%, *p* < 0.05) (Figure 3B).

Figure 4 shows a color-coded summary of the significant changes occurring in the relative power of the different frequency bands for all the genotypes. At three months, all the tg lines showed a decrease in the relative SO power in *sr-lm* (1500 μm), but only B6.152H mice reached statistical significance (−42%, *p* < 0.05).

In six-month-old B6.152H mice, with increased Aβ accumulation and plaque deposition, the loss of relative SO power was larger (−57% *sr-lm*, *p* < 0.01) and invaded the DG (−47% *lo-sg*, *p* < 0.05, Figure 4), with a marked reduction in SO/delta power ratio (Figure 3B) that reached 70% in the DG (*lo-sg*, *p* < 0.05). In these mice, there was also an imbalance between Low (0.1–4.7 Hz) and High (4.7–190 Hz) frequency ranges (Figure 3C and Figure 4), which can also be appreciated from the normalized PSD plots at *sr-lm* (Appendix A). Upon quantification, the Low/High power ratio was decreased by more than 50% in both DG (*p* < 0.01) and *sr-lm* (*p* < 0.05, Figure 3C). In particular, when compared to age-matched WT mice, we observed a significant and large increase in the relative power of theta band in *sr-lm* (+63% *p* < 0.01), and beta band in the whole DG (+70% *sg*, *p* < 0.01) (Figure 4).

Concerning higher frequencies, there was also a significant increase in the relative epsilon power that emerged at the hippocampal level of three- (PS2.30H and B6.152H) and six-month-old (APPSwe) mice, whereas PS2KO mice showed a decrease in the same frequency band in the DG (Figure 4). In summary, at the CA1 level, a dampened SO/delta ratio, coupled with an increase in theta and beta bands, makes it possible to identify a specific and selective marker of AD progression in B6.152H mice, whereas an increase in the epsilon relative power is a common feature of APPSwe, PS2.30H and B6.152H mice which is not shared with PS2KO ones.

### 3.3. Altered SO Functional Connectivity in Predeposition B6.152H Mice

To identify alterations in resting functional connectivity in young, asymptomatic AD mice, we used the cross-correlation of the “instantaneous amplitudes”, i.e., the amplitudes of the complex Hilbert transform of LFPs, as previously described [43] and summarized in Appendix A. Figure 5A shows matrices reporting the maximal cross-correlation coefficients for instantaneous SO amplitudes (upper-right) and corresponding latencies (lower-left) for three-month-old WT mice. Of the four age-matched genotypes, only B6.152H mice showed marked and significant changes in both cross-correlation coefficients and latencies (Figure 5B, black lines and Appendix A).

When considering lags in control, three-month-old WT mice, SO in L4/6 anticipated those in DG (Figure 5A), whereas this anticipation was significantly reduced in age-matched B6.152H mice (Figure 5B, lower-right, black lines). In these mice, the loss of cross-correlation was statistically significant within the HPF and between the *sr-lm* and the whole PPC (Figure 5B, upper-right, black lines). To quantify these changes at a large regional level, we averaged the cross-correlation coefficients and latencies from adjacent channels that correspond to defined hippocampal and cortical regions (Figure 6A). For SO, the temporal lag between L4/6 and DG changed from −68.1 ± 8.9 ms in WT mice to −5.7 ± 16.1 ms in B6.152H mice (mean ± SEM *p* < 0.05). SO in L2/3 were also about 40 ms late with respect to those in DG (*p* < 0.05) (Figure 6B), thus confirming a cortico-hippocampal delay for SO in B6.152H mice. Notably, for all the other genotypes, at three months of age, matrices of cross-correlation and latency resembled those of WT mice (Appendix A).

In B6.152H mice, loss of cross-correlation and latency for SO were observed also at six months of age (Figure 5D), yet no statistical significance was detected at the regional level with respect to age-matched WT mice (Figure 5C,D and Appendix A). This finding can be explained by the fact that six-month-old WT mice also showed reduced cross-correlation in the DG and loss of latency in L4/6 when compared to young mice (Figure 5C, black lines). No similar changes in SO connectivity were found in the other tg lines, except for six-month-old APPSwe mice, that showed a significant reduction in latency between L2/3 and DG, similarly to what found in young B6.152H mice (Appendix A). Altogether, in the PPC of young B6.152H mice, SO propagation was significantly delayed with respect to SO in DG. Of note, in these mice, the loss of SO connectivity was also accompanied by changes in delta connectivity, albeit less pronounced, with a significant increase in cross-correlation between L2/3 and CA1, and a reduced lag in CA1 (Appendix A). Similar effects on delta cross-correlation and latency were absent in all the other genotypes (Appendix A).

### 3.4. Loss of Cortico-Hippocampal Phase-Amplitude Coupling in PS2-Based AD Mice

During SWS, the cerebral cortex coordinates the time of transmissions from the hippocampus via propagated low frequency signals [22]. A relevant feature of coordinated brain activity is the presence of nested oscillations, emerging when slower rhythms influence faster ones in a dynamic fashion [44,45]. We wondered whether the cross-frequency coupling (CFC) between cortical SO and hippocampal higher frequencies was also affected. We quantified Phase-Amplitude Coupling (PAC) by computing the General Linear Model (GLM) index for SO with respect to higher frequencies, as described in Appendix A. When compared to age-matched WT animals, three- and six-month-old PS2.30H mice showed a significant reduction of PAC between SO in cortical L4/5 and fast gamma (FG, 45–90 Hz) in CA1 (*sr-lm*). The same effect was found in six-month-old B6.152H and APPSwe, but not PS2KO, mice (Figure 7A).

A specific signature of B6.152H mice was the progressive and dramatic loss of PAC between SO and higher frequency bands, observed at the different depths (intraregional PAC) (Figure 7B). In three-month-old B6.152H mice, at the hippocampal level, more precisely in *sr-lm* and *upper-sg*, SO PAC was attenuated with gamma and epsilon bands, respectively (Figure 7B). Notably, at six months of age, alterations in SO PAC invaded not only the entire HPF, but also the PPC (Figure 7B), an area also affected by Aβ deposition and gliosis. At this age, SO PAC with beta band was also impaired (Figure 7B). Interestingly, within restricted areas of the HPF (*lower-sg*) and PPC (L4/5), SO PAC with gamma bands was reduced also in young PS2.30H but not APPSwe and PS2KO mice (Figure 7B). APPSwe mice mimicked B6.152H only at six months, with attenuation of SO PAC with beta and gamma bands observed only in L4/5 (Figure 7B).

In the regions where the SO cortico-hippocampal PAC was affected, we did not find significant changes in the PAC between cortical delta (1.7–4.7 Hz) waves and the higher frequencies (not shown). In contrast, this type of PAC was significantly reduced at the intrahippocampal level of AD (B6.152H, PS2.30H and APPSwe) but not PS2KO mice, especially in the coupling with the FG/epsilon (45–190 Hz) bands (Appendix A). It is worth noting that, in humans, for successful encoding at the hippocampal level, the PAC with gamma (34–130 Hz) frequencies relies on the slow-theta (2.5–5 Hz) rather than the classical theta (5–10 Hz) band-i.e., on a frequency range that corresponds to our delta [46].

### 3.5. α- and β-APP Carboxy-Terminal Fragments are Not Involved in Reduced SO Connectivity

AD mice that overexpress human APP mutations accumulate soluble Aβ peptides according to a time course that depends on different factors, among which the most relevant is the coexpression of PS1 or PS2 FAD-linked mutations that accelerate the process. Alterations of brain network activity and behavior have also been linked to APP overexpression or to the accumulation of its proteolytic products [47,48,49,50]. In particular, the β-carboxy-terminal fragments (β-CTFs) derived by the β-secretase activity have been implicated in hyperexcitability [47,48,49,50]. Western Blot (WB) analysis of cortical homogenates showed that only six-month-old B6.152H mice showed significantly higher levels of α- and β-CTFs, i.e., six and three times higher, respectively, compared to three-month-old mice (Figure 8C,D), but they were below threshold in six-month-old WT, PS2.30H and PS2KO mice (Figure 8B). Of note, at three months, B6.152H and APPSwe mice showed similarly low levels of both CTFs (Figure 8C,D). These findings suggest that CTFs are not likely responsible for both the imbalance in SO/delta power and the loss of SO connectivity that characterize three-month-old B6.152H, but not APPSwe mice. Of note, these network features are not simply due to APP overexpression, since three-month-old APPSwe and B6.152H have similar expression levels of full-length APP (APPSwe: 100 ± 11.5%; B6.152H: 116.07 ± 21.7%, mean ± SEM, *n* = 15 samples from six mice for each tg line).

## 4. Discussion

The B6.152H line is a double tg AD mouse model that accumulates amyloid plaques and displays gliosis starting at six months of age, i.e., much earlier than the single tg APPSwe line [12,26,27,51,52]. In addition, the double tg line shows defects in neuronal/astrocytic Ca^2+^ homeostasis and endoplasmic reticulum (ER)-mitochondria tethering, shared with the PS2.30H line, which, conversely, lacks amyloidosis and gliosis [12,32].

Our intent was to identify, in anesthetized AD mice, brain network changes that anticipate the onset of amyloidosis, proceed with amyloid accumulation, but are distinct from those linked to the expression of either the PS2 or APP mutations. The rationale of the approach was the following: early alterations in spontaneous brain activity that appear only in young B6.152H mice are more likely to be linked to Aβ oligomers that start to accumulate at three months of age, first intraneuronally and then extracellularly, whereas network alterations, observed in these mice at six months of age, can be correlated to plaque deposition and gliosis; in addition, changes not linked to Aβ load, but due to the expression of the human APP-Swedish or PS2-N141I mutation, should be present also in the single PS2.30H and APPSwe lines. Finally, a comparison with PS2KO mice allowed us to investigate circuit defects associated with the loss of function of PS2, an additional aspect linked to neurodegeneration [34]. A summary of the major changes found in the different tg lines is shown in Figure 9.

### 4.1. Loss of SO Connectivity Distinguishes Predeposition B6.152H Mice

The most striking and precocious marker, exclusively found in young B6.152H mice, derives from the analysis of cortico-hippocampal connectivity. In particular, we identified an early loss of the resting brain connectivity that is based on SO. Changes in SO connectivity selectively appeared in these mice at the onset of Aβ accumulation, but were not found in the single tg lines or in the PS2KO mice. Loss of SO cross-correlation was found not only within the DG and CA1, but also between these two regions. Moreover, all the PPC layers significantly showed alterations in cross-correlation and latency for SO with respect to the DG and *sr-lm,* two major hippocampal inputs from the LEC. An Aβ-dependent impairment of slow-wave propagation and coherence was also found by measuring spontaneous Ca^2+^ activity across the neocortex and the HPF of anesthetized AD mice [23]. Notably, an increase in latency was also found in APPSwe mice, but only at six months of age. Our findings are also consistent with data obtained in humans, reporting a direct correlation between Aβ accumulation and disruption of slow waves during non-rapid eye movement (NREM) [53]. Interestingly, in young B6.152H mice, we also observed a significant increase in delta connectivity, as measured by the cross-correlation coefficients between L2/3 and CA1. A competition between SO and delta waves was recently suggested by Ganguly and coworkers, where the balance between SO and delta waves dictates the ratio between memory consolidation and forgetting in postlearning sleep [54].

### 4.2. Expression of the PS2 Mutation Causes Loss of SO PAC Independently of Amyloidosis

In encoding and memory recovery, an important feature is the nesting between low and high frequency bands, a phenomenon measured by CFC [38,44,45]. In particular, cortico-hippocampal PAC for slow-waves is relevant to memory consolidation, both at rest and during SWS [22,39,55]. From cross-regional analysis, we observed a significant loss of PAC between SO in L4/5 and FG in CA1 (*sr-lm*). Interestingly, attenuation of cortico-hippocampal PAC is an early feature of three-month-old PS2.30H, but not PS2KO mice. Furthermore, in young mice, loss of intraregional SO PAC was observed only in PS2-based mice (PS2.30H and B6.152H), highlighting the primary role played by the PS2 mutation in SO coupling. Dampened PAC in the SO range might also involve Ca^2+^ dysregulation in glial cells, given that the PS2-but not the APP-mutant is expressed also at this level. Indeed, in astrocytes, a Ca^2+^-based excitability controls the UP states of surrounding neurons [56] and contributes to both fast dynamics of neural circuits [57,58,59] and memory coding [60,61].

### 4.3. Amyloidosis Worsens the Loss of SO PAC

An impairment of cortico-hippocampal SO PAC was also present in six-month-old APPSwe and B6.152H mice, yet it was much more pronounced in these latter. Furthermore, only in these mice, we found a dramatic loss of intraregional SO PAC with higher frequencies, from beta to epsilon, at both the HPF and PPC level. Taken together, these findings suggest that the loss of SO PAC has both Aβ-dependent and -independent components that synergize in B6.152H mice.

### 4.4. Low/High Frequency Power Imbalances Mark the Progression of Amyloidosis in B6.152H Mice

When we considered the relative contribution of the different frequency bands to the total power, we noticed that, among the young cohorts, only the B6.152H mice showed a large and significant reduction in SO in CA1 (*sr-lm*). This was also associated with a marked reduction in the power ratio between SO and delta band. These changes worsened in six-month-old B6.152H mice, invading the DG. At six months, there was also a further, significant power imbalance between Low (1–4.7 Hz) and High (4.7–190 Hz) frequency ranges, with the reduction in SO/delta power ratio being associated with significant increases in theta and beta relative powers, which covered the whole HPF; this was not found in the other tg lines. In contrast, significant power defects were not found in the PPC of three- and six-month-old B6.152H mice, yet this region is precociously involved in the loss of SO connectivity, in terms of both cross-correlation and delay with respect to the HPF, as well as of cortico-hippocampal SO PAC. The PPC is part of the DMN and, as an association area, is functionally connected to the HPF, mainly through the EC. It is also considered to be among the most vulnerable area to Aβ toxicity, even in the absence of plaques [62,63]. It remains an open question whether there is a causal relationship between Low and High frequency imbalances; however, a vicious cycle might be part of these network changes [23]. Long-range effects of Aβ on brain connectivity have recently been suggested in a rat AD model, in which distant Aβ induces regional metabolic vulnerability and greater susceptibility to local Aβ [5].

It is worthy of note that by employing a voltage-sensitive dye, reduction in slow-wave correlation and power was similarly found in the somato-sensory cortex of predeposition APPSwePS1dE9 mice [24]. In addition, the local application of Aβ-oligomers mimicked slow-wave alterations in control mice, while chronic optogenetic stimulation, in the same frequency range, reduced plaque deposition and Ca^2+^ imbalances [24]. In human studies by means of electroencephalography (EEG), reduction in slow-wave (0.6–1 Hz) activity during sleep was related to higher cortical Aβ burden, and has been suggested as a potential biomarker of AD progression [55,64].

### 4.5. Molecular and Cellular Pathways Involved in SO Changes

In B6.152H mice, alterations in the SO range were found for both power and connectivity, as well for the coupling with higher frequencies. All these features emerged early in the HPF, also invading the PPC with the advance of amyloidosis. From a molecular point of view, these defects are likely caused by the initial surge of intraneuronal Aβ accumulation and by the late extracellular Aβ deposition. Nonetheless, a comparison with the PS2.30H line that does not accumulate Aβ highlights Aβ-independent effects, which are linked to the PS2 mutation [65]. Moreover, their absence in PS2KO mice suggests that PS2.30H mice are dominated by gain effects rather than loss-of-function, contrary to what was previously suggested [34]. Regarding the molecular aspects linked to the PS2 mutation, we are aware of alterations in Ca^2+^ homeostasis and metabolic pathways, as well as in autophagic steps, all of which are relevant issues from a pathogenic perspective of AD [32,65,66,67,68,69,70,71]. In particular, we cannot exclude that some of the observed defects are linked to alterations in Notch signaling, even if Notch is a major substrate of PS1 rather than PS2 [34].

Concerning α,β-CTFs, we can exclude the possibility that they are involved in the disruption of SO connectivity and power imbalance of three-month-old B6.152H mice, since these effects are absent in three- and six-month-old APPSwe that show similar levels of both CTFs and full-length APP. Yet, we cannot exclude that they are involved in the network decay of six-month-old B6.152H mice, that show high level of both CTFs. Indeed, other APP proteolytic products which were not quantified in this work might contribute to the network changes found in B6.152H mice.

In PS2KO mice, we also found no evidence that PS2 loss-of-function effects clearly contributed to the changes in SO power, functional connectivity, and PAC described in PS2-based AD mice.

From a cellular point of view, these findings suggest that early Aβ accumulation might impair the intrinsic generator of SO in L4/5 [72,73,74,75], since in young B6.152H mice, L4/5 is the most affected area when considering the loss of cortico-hippocampal latency for SO. Our data are also consistent with the early disruption of cortical L5 connectivity found in 5xAD mice [76] and with the general view of AD as a disconnection syndrome [77]. In young B6.152H mice, SO also lose correlation within the DG and CA1, likely by deficits occurring in those interneurons that are phase-locked to slow waves in the PPC [78]. Indeed, the major cortical inputs from the LEC reach the HPF in *sm* and *sr-lm*, the two areas that appear primarily affected in PS2-based AD mice, both in terms of power imbalances and connectivity. This is also consistent with loss of LEC functionality in the early stages of AD [6,8]. Finally, the PS2 mutation alone is sufficient to impair the coupling between the phase of SO in L4/5 and the amplitude of higher frequencies in CA1, in the absence of amyloidosis, suggesting that perturbed ER-mitochondria crosstalk is part of cortical AD vulnerability.

### 4.6. Study Limitations

Our study provides evidence for cortico-hippocampal deficits in SO synchronization in mice under anesthesia, a condition where the Low (<4.7 Hz) frequencies largely dominate the power spectrum (about 70% of the total power), thus possibly emphasizing alterations in this range. Given that anesthetized and sleeping animals share many similarities, and differences [79], our conclusions should be confirmed by recordings during SWS.

In mice, EEG studies show that ketamine anesthesia decreases the theta (3–12 Hz) and enhances the gamma power (30–80 Hz) of the background network activity [80], possibly introducing a confounding factor when comparing studies with different anesthesia protocols. We should also mention that a potential interaction between the anesthesia and the disease could eventually occur, although we did not observe differences in the required anesthesia doses. Moreover, for the analyses, we chose only those windows with stable heart and respiration rates according to predefined ranges. All the animals were thus recorded under the same conditions, and the observed differences were likely due to changes in the network.

The use of homozygous tg mice is a further limitation; however, employing mouse lines carrying the same mutations and promoters, in a similar background (>95% C57BL/6J), limits artifacts due to the latter or to genetic drift, and confirms the role played by PS2 in two independent tg lines. Furthermore, applying a multiple comparison approach reduces the incidence of false positives. Of note, three mouse lines, i.e., B6.152H, PS2.30H and APPSwe, share similarities in different aspects of the resting network activity, such as increased power contribution in the epsilon band and loss of SO PAC at six months of age. In contrast, the PS2KO line appears rather different from the other tg lines. Yet, one similarity emerged in SO connectivity: six-month-old PS2.30H and PS2KO mice showed a significant increase in SO cross-correlation between L4/6 and upper regions of CA1 (*sp and so*) (see black lines in Appendix A), albeit one that was not present at a macro-regional level (Appendix A). Hitherto, we cannot exclude that further analyses or a less conservative approach could reveal network properties linked to a loss of PS2 function.

Finally, this study was limited by a lack of behavioral data, answering the question of whether the observed network dysfunctions would affect mouse behavior. Curiously, in the original PS2APP mice, obtained by crossing single tg lines, signs of impaired Y maze behavior were found at four months, before the defective water maze learning that occurred only at eight months, after plaque deposition [27].

## 5. Conclusions

By recording LFP signals under anesthesia, we identified specific brain network alterations of spontaneous electrical activity that occur in B6.152H mice at the start of Aβ production and during plaque deposition and gliosis. Impaired SO cortico-hippocampal connectivity marks the onset of amyloidosis, whereas power imbalances in the Low–High frequency range and loss of SO coupling to higher frequencies mark the progression of the disease. Both Aβ-dependent and -independent insults—the latter also being linked to the PS2 mutation—contribute to the diseased phenotype. Overall, our findings identify PS2.30H and B6.152H mice as suitable and reliable models to follow brain alterations linked to AD, in a context where the cell defects due to the PS2 mutation—primarily Ca^2+^ dysregulation—might play a precocious role in the pathogenesis of AD.

## Figures and Tables

**Figure 1 cells-09-00054-f001:**
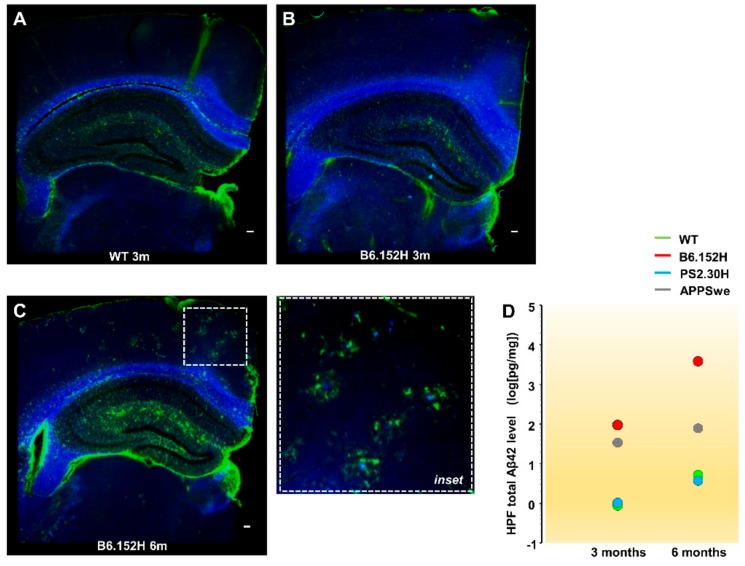
Amyloidosis and gliosis in B6.152H mice. Representative images of coronal sections (50 μm) from the left hemibrains of WT (**A**) and B6.152H (**B**,**C**) mice. At the hippocampal and cortical levels, six- (**C**) but not three- (**B**) month-old B6.152H mice show plaques, stained with methoxy-X04 (blue) surrounded by activated astrocytes, stained by GFAP antibody (green); see also the *inset* (scale bar = 0.1 mm). (**D**) Total Aβ42 accumulation, estimated with a human/rat Aβ42 ELISA in the HPF of three- (WT: 0.6 ± 0.5; PS2.30H: 0.7 ± 0.3; APPSwe: 16 ± 1.6; B6.152H: 110 ± 52 pg/mg wet tissue) and six-month-old mice (WT: 2.3 ± 0.2; PS2.30H: 3.5 ± 2.4; APPSwe: 92 ± 46; B6.152H: 4450 ± 70 pg/mg wet tissue), mean ± SEM, *n* = three mice for each age/genotype.

**Figure 2 cells-09-00054-f002:**
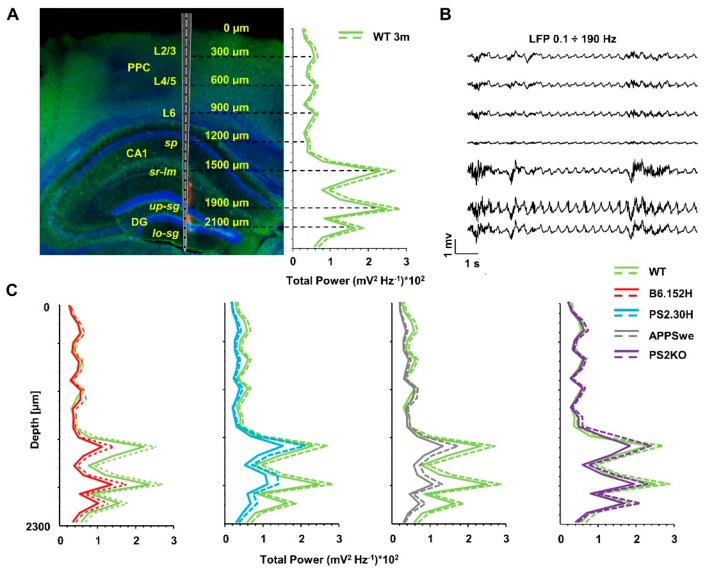
Simultaneous recording of LFP signals from PPC and dorsal HPF. (**A**) *Left panel*: Coronal brain section (70 μm) from a three-month-old WT mouse visualized by Hoechst-33342 (blue), DiI (red) and bright field (green) signals. The image also shows a schematic drawing of the linear probe, the total power profile recorded by the 24 channels (mean + SEM), and the depths of the seven channels used for analysis. (**B**) Representative band-pass (0.1–190 Hz) filtered LFP recordings from the selected channels. (**C**) Average total power profile at three months of each tg line compared to that of age-matched WT mice (mean + SEM).

**Figure 3 cells-09-00054-f003:**
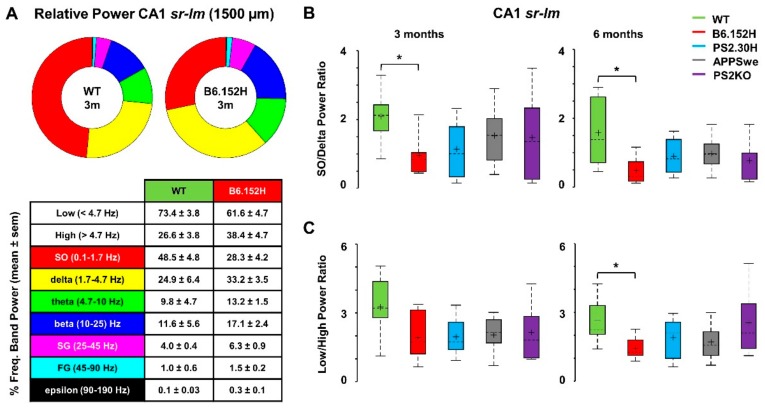
Hippocampal power imbalances characterize B6.152H mice. Ring charts of the relative power distribution in the different frequency bands in CA1 (*sr-lm*) of three-month-old WT and B6.152H mice (**A**), top panel); relative power at each frequency band expressed as percentage of total power (mean ± SEM) (**A**), bottom panel). Boxplots of the power ratios between SO (0.1–1.7 Hz) and delta (1.7–4.7 Hz) bands (**B**), and between Low (0.1–4.7 Hz) and High (4.7–190 Hz) frequency bands (**C**), at three and six months of age, * *p* < 0.05.

**Figure 4 cells-09-00054-f004:**
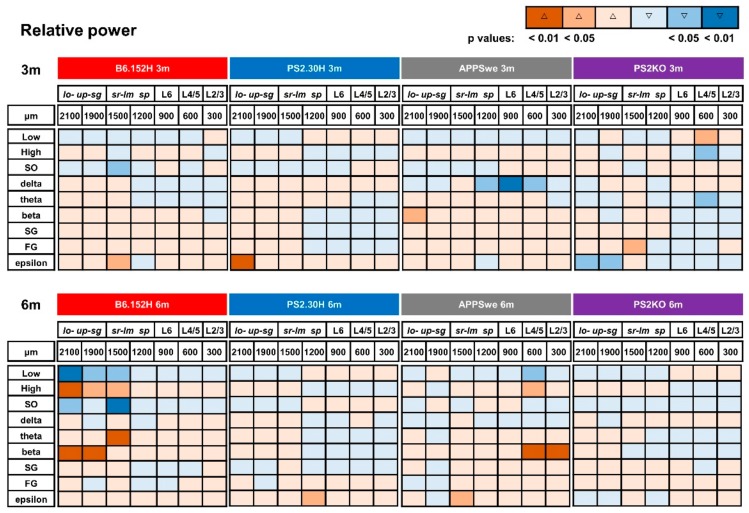
Age-matched genotype comparison of changes in the relative band powers. Synoptic view of the changes occurring in the relative band powers at the different depths of three- (upper panel) and six- (lower panel) month-old tg mice with respect to age-matched WT mice. Changes are color-coded in terms of increase (Δ) or decrease (▽) and statistical significance.

**Figure 5 cells-09-00054-f005:**
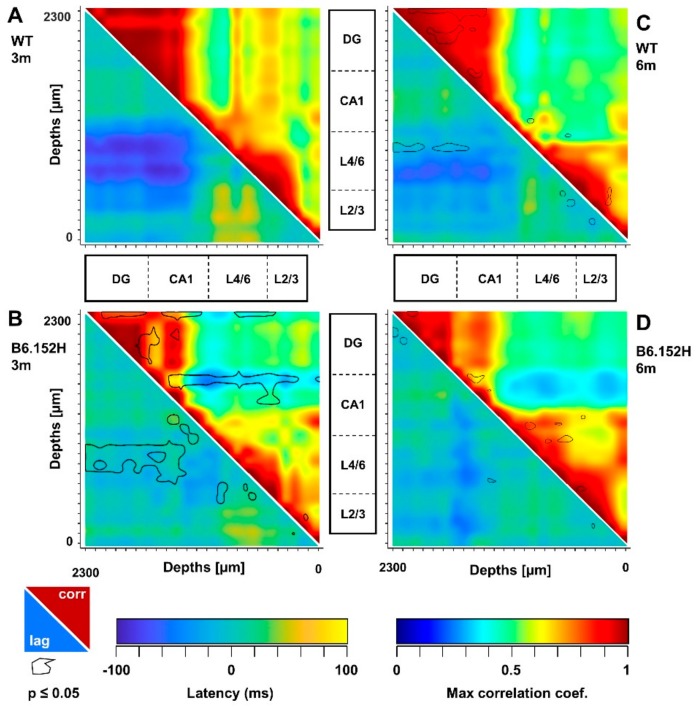
Reduced SO cortico-hippocampal connectivity in young B6.152H mice. SO connectivity was measured in terms of maximal cross-correlation coefficients and latencies of the instantaneous SO amplitude, as described in Appendix A. Matrices of cross-correlation coefficients (upper-right) and latencies (lower-left) for WT (**A**,**C**) and B6.152H (**B**,**D**) mice were obtained by comparing each recording channel with all the other channels. The B6.152H matrices report the areas with significant changes with respect to age-matched WT mice (*p* < 0.05, black line). In the six-month-old WT matrices (**C**), the comparison is with three-month-old WT mice in panel A.

**Figure 6 cells-09-00054-f006:**
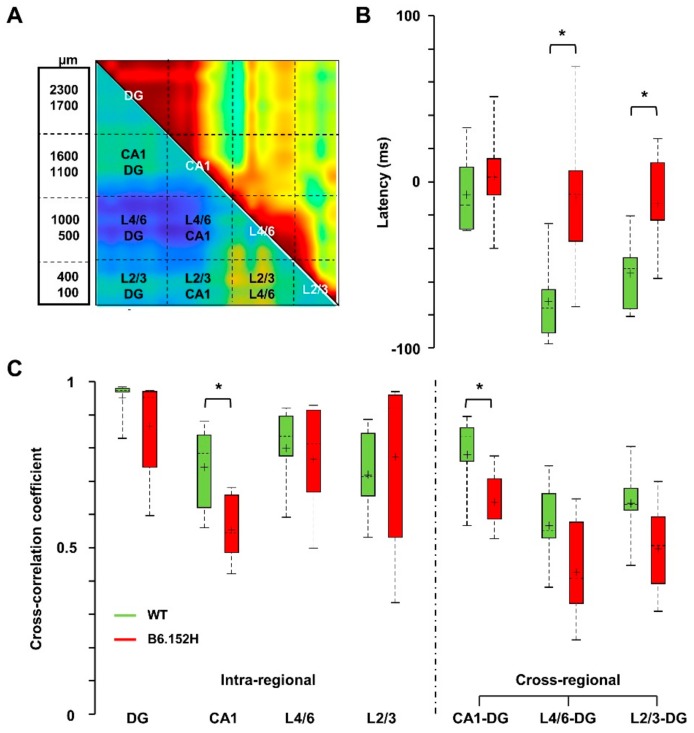
Loss of SO cortico-hippocampal correlation and latency in young B6.152H mice. For quantitative analyses of regional changes, SO maximal cross-correlation coefficients and latencies of each mouse were averaged within (intraregional) and between (cross-regional) regions, according to the scheme overlaid to the matrices of the three-month-old WT mice (**A**). Values were then averaged by age and genotype and shown in boxplots as cross-regional latencies (**B**) and intra- and cross-regional cross-correlation coefficients (**C**), * *p* < 0.05.

**Figure 7 cells-09-00054-f007:**
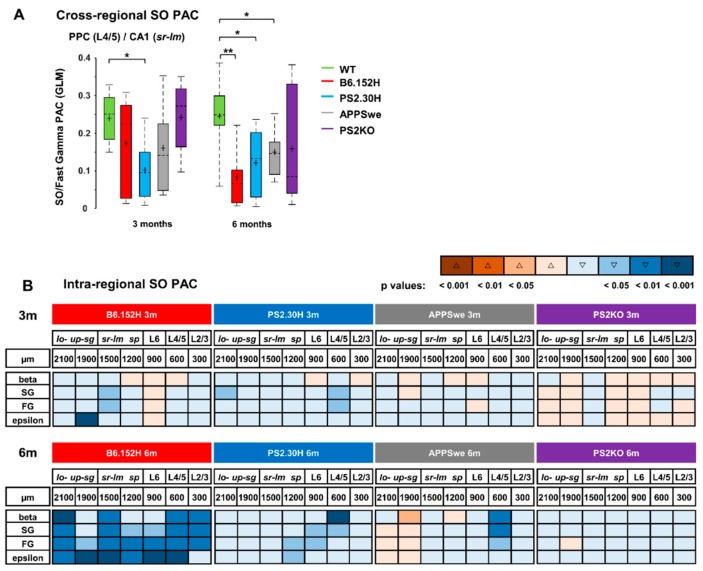
Reduced SO Phase-Amplitude Coupling in AD mice. PAC between SO and higher frequencies was measured by the GLM index as described in Appendix A. The cross-regional PAC was measured between SO in PPC and higher frequencies in the HPF. Significant impairment was found between SO in L4/5 and FG in CA1 (*sr-lm*) (**A**), ** *p* < 0.01; * *p* < 0.05.The intraregional PAC was measured between SO and higher frequencies at each depth (**B**). Changes were color-coded in terms of increase (Δ) or decrease (▽) and statistical significance with respect to three- (upper panel) and six- (lower panel) month-old WT mice.

**Figure 8 cells-09-00054-f008:**
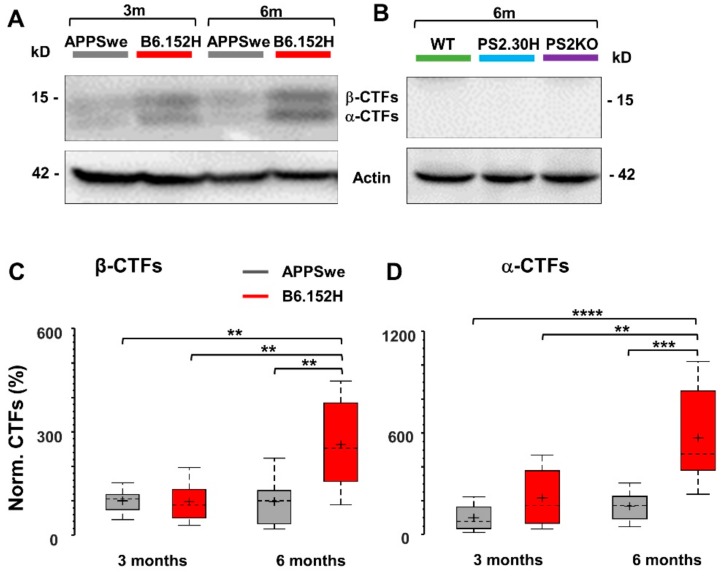
Expression levels of APP-CTFs. Right cortices obtained from three- and six-month-old WT and tg mice were assayed with SDS-PAGE and WB, as described in Appendix A. Representative WB of APP-CTFs, assayed with an anti-APP-CTF (CT695), from APPSwe and B6.152H (**A**) and WT, PS2.30H and PS2KO (**B**) mice. In panel B, even at higher exposure, the CTFs were below threshold. Boxplots of α-(**C**) and β-(**D**) CTFs (*n* = 15 samples from six mice per age and genotype); note the different scales in panels C and D. Values were normalized to actin and expressed as the percentage of the average value measured in three-month-old APPSwe mice (non-parametric Kruskal-Wallis test with the Dunn-Sydak correction for multiple comparisons, **** *p* < 0.0001, *** *p* < 0.001, ** *p* < 0.01).

**Figure 9 cells-09-00054-f009:**
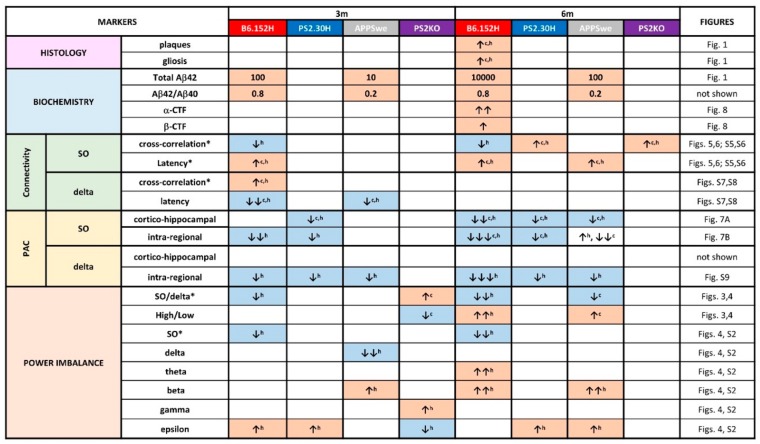
Summary of histological and electrophysiological changes in the different tg lines. Changes are color-coded as defined in Figure 4; empty boxes indicate no significant changes with respect to age-matched WT mice; c, cortex; h, hippocampus; * parameters changed only in young B6.152H mice. Number of arrows is a rough indication of the entity of change/extension of the region involved, as quantified in the respective figures. For histology and biochemistry, data are also from Refs. 12, 26, 33. Total Aβ42 levels are expressed in pg/mg wet tissue and approximated for comparison.

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
