# Peer review of "Dampened Slow Oscillation Connectivity Anticipates Amyloid Deposition in the PS2APP Mouse Model of Alzheimer’s Disease"

_cells, 2019, doi:10.3390/cells9010054_

Round 1
Reviewer 1 Report
This work investigated the brain network dysfunctions in Alzheimer's disease (AD) using the AD mouse models from 3 and 6 months of age of B6.152H, PS2.30H and APPSwe mices (double transgenic B6.152H mice that express the human APP-Swedish and PS2-N141I mutations as well as wild-type and single transgenic PS2.30H and APPSwe mice to studying the specific contributions of PS2 and APP mutations) by recording the local field potentials (LFP) signals in the posterior parietal cortex (PPC) and dorsal hippocampal formation (HPF) under anesthesia. The differences related to either the initial accumulation of the beta-amyloid or to the deposition of plaques and gliosis were also investigated. Behavioral study would be interesting to perform! Although transgenic mouse lines, based on human mutations linked to familial AD (FAD), cannot mimic the full spectrum of the human AD and the sporadic AD (SAD) accounts for ~99% of AD cases, however, in general, I find this study interesting and recommend the publication of this manuscript.
Author Response
Reviewer 1:
Thank you for the positive evaluation of our work.
…. Behavioral study would be interesting to perform!...
We know that the lack of behavioral studies can be seen as a limitation of the present work, and we have now included this point in the Discussion (see page 24, end of paragraph 4.6). Indeed, previous data obtained with the original PS2APP model, observed a behavioral deficit occurring as early as 4 months of age in the Y maze test (Richards et al. ’03). Preliminary data obtained in the B6.152H line confirm this finding. They are part of a larger investigation, not yet completed, done by a group with whom we are now collaborating on this relevant issue.

Reviewer 2 Report
This manuscript investigated spontaneous electrical activity in the hippocampus of mouse models with human presenilin-2 (PS2) and/or amyloid precursor protein (APP) mutations. They found that cortico-hippocampal connectivity of slow oscillations (SO) and hippocampal power ratio between SO and delta waves were altered in PS2APP-Tg mice at 3 months of age. At 6 months, a loss of cortico-hippocampal phase-amplitude coupling (PAC) between SO and higher frequencies was detected in PS2APP-Tg and PS2-Tg mice. Although one major weakness is that the electrical activity was measured under anesthesia condition in those mice, their findings are potentially interesting. However, experimental procedures are insufficiently described, in particular for biochemical and histological analyses. Additional concerns are as follow.
Although references are cited, the brain region in which the electrode was inserted should be clarified in methods (Lengths from bregma and angle etc.). There is no result assessing whether thickness of cortex or hippocampus is same in those Tg mice at 3 and 6 months old. Fig 1: The methods to measure Aβ is not described. Since human and murine Aβ are quite different, it might be difficult to analyze in same procedures. In Fig 1D, “plaques” and “no plaques” should be removed, because quantitative assessments for plaque burden were not performed. Authors should investigate if human Aβ40 and Aβ42 levels are associated with electrical activities in PS2APP-Tg and APP-Tg mice. Fig 8: It is unclear which antibody was used to measure CTFs. It would be more informative if CTFα, CTFβ and AICD are quantified, and their associations with electrical activities are examined. Notch signaling may be changed by PS2 overexpression, which possibly contributes to the results.Author Response
Reviewer 2:
We thank the reviewer for the valuable suggestions.
...Although references are cited, the brain region in which the electrode was inserted should be clarified in methods (Lengths from bregma and angle etc.).
Concerning probe insertion, as specified in Materials and Methods (page 6, line 4 from top) …
To locate the DG for probe insertion, we used landmarks on the skull of each mouse, according to the procedure described by Huang et al. [36]. Following this approach, we could not specify single set of coordinates. We agree with the Reviewer that these data are important and we have added Supplementary Fig. 1A (see also page 9, bottom), describing the procedure for coordinate location and reporting the average values obtained in WT and B6.152H, the two groups that mostly differ. In Supplementary Fig. 1B, we have also summarized the anesthesia conditions and reported the average bpm and rpm of the same groups. No statistically significant difference was found between B6.152H and age-matched WT mice, suggesting that a similar anesthesia level was reached in the time windows we selected for the analyses, according to the procedure described in Materials and Methods.
…There is no result assessing whether thickness of cortex or hippocampus is same in those Tg mice at 3 and 6 months old…
The reviewer is right, we don’t have this type of information. We based our findings on the comparison of the total power profile. It should also be taken into consideration that the 24 electrodes are 100-mm apart, with each electrode collecting signals from an area of about 200-mm diameter; thus, there is substantial overlap between signals recorded by adjacent electrodes and only mistakes in probe location much larger than 200-mm should have significantly affected the power profile.
To our knowledge, there are no substantial changes in cortical or hippocampal thickness in B6.152H mice and neurodegeneration has not been reported for these mice, between 3 and 6 months of age. However, we cannot completely exclude that subtle changes occurring at the hippocampal level may affect the network behavior.
… The methods to measure Aβ is not described. Since human and murine Aβ are quite different, it might be difficult to analyze in same procedures…
The methods to measure Ab42 were fully described in Supplementary Materials. Unfortunately, we had not mentioned this piece of information in the Results section; it is now included in the legend for Fig. 1.
We agree with the Reviewer that human and murine Aβ are quite different. For that reason, for the results shown in Fig. 1D, we employed the WAKO assay, the only one that guarantees the dosage of both human and rat (mouse) Ab42. This assay is suitable - especially the high-sensitive one - to evaluate the endogenous Ab42 levels in WT and PS2.30H mice. We also mentioned the use of the human Ab42 Millipore assay that reliably estimates Ab42 levels in APPSwe and B6.152H mice, giving results similar to those found with the WAKO assay. These details were present in the original version of Supplementary Materials: in particular, the characteristics of the ELISA assay, i.e, its capability to detect both human and rat/mouse Ab42.
… In Fig 1D, “plaques” and “no plaques” should be removed, because quantitative assessments for plaque burden were not performed…
The Reviewer is right, we have changed Fig. 1D accordingly, given that in this work there was no quantitative assessment for plaque burden. Since this type of analysis was extensively carried out by our group (Fontana et al. ’17) as well by others (Richards et al. ’03; Ozmen et al. ’09); we have included this type of information in the Summary shown in Fig. 9. We have also expanded the Results section to better explain Fig. 1D and allow the comparison between APPSwe and B6.152H mice (page 8, line 9 from bottom).
… Authors should investigate if human Aβ40 and Aβ42 levels are associated with electrical activities in PS2APP-Tg and APP-Tg mice…
This is a very interesting issue, albeit it is rather difficult to obtain a proper quantification of Ab levels in the exact areas used for the recording sites, i.e., specifically part of the hippocampus (DG and CA1) and the PPC. Since this was not planned from the beginning of the work, we do not have the possibility to properly answer this question (lack of tissues from the recorded mice, that was also used for WB). We cannot exclude that estimating the level of Ab42 oligomers - rather than total or SDS soluble Ab42 - in a larger number of mice, might better address this issue. Yet, as mentioned in the Discussion, we should also consider that Ab accumulation can affect distant, Ab-free regions, through connectivity (Pascoal et al. 2019) (page 23, line 2 from top).
… Fig 8: It is unclear which antibody was used to measure CTFs. It would be more informative if CTFα, CTFβ and AICD are quantified, and their associations with electrical activities are examined…
The type of antibody used for WB was specified in the Supplementary Materials (anti-APP-CTF, CT695, Invitrogen). We agree that having the Materials and Methods split into two parts worsens the readability of the manuscript, but this is an editorial constraint. For that reason, this important piece of information has now been included in the legend for Fig. 8.
As far as CTFs are concerned, we carried out additional experiments to improve the statistics and, as suggested, we quantified separately the two bands, corresponding approximately to α- and β-CTFs. New panels were included in Fig. 8 and these findings were presented in paragraph 3.5. None of the two types of fragments was significantly increased at 3 months in both APPSwe and B6.152H mice, suggesting that they were likely not involved in the network changes occurring at the beginning of the amyloidosis. However, β-CTFs and especially α-CTFs were largely increased in 6-month-old B6.152H but not APPSwe mice, possibly contributing to the network imbalances (see Discussion, paragraph 4.5, page 23, line 4 from bottom). Given that detection of increased levels of CTFs might be limited by the choice of the antibody or SDS-PAGE, we removed the sentence on CTFs from the Abstract, as a major statement, thus better focusing the Abstract; this is also in line with the requirements from Reviewer 4.
… Notch signaling may be changed by PS2 overexpression, which possibly contributes to the results...
Yes, this is an additional possibility, that we did not mention, among the Aβ-independent effects of the mutant PS2. Unfortunately, we did not have the possibility to check whether the level of Notch and its proteolytic products were changed. We have now mentioned this issue in the Discussion (page 23, line 6 from bottom). Indeed, a plethora of other players are likely involved in the observed changes. Our aim was to highlight the primary role played by the mutant PS2, not mimicked by the loss-of-function PS2KO mice.

Reviewer 3 Report
In this study, Leparulo and colleagues report on changes in coordinated slow wave activity in early-stage mouse models of AD. For that, they use LFP recordings using linear multielectrode probes in hippocampus and posterior-parietal cortex. They find an age-dependent and model-dependent dysregulation of slow wave coherence between the two recording regions, and, maybe most interestingly, also an impairment of the coupling of slow oscillations with higher rhythms such as gamma, which is instrumental for memory consolidation. This study is of potential high interest, as it focused on long-range connectivity in a specific brain state, an aspect often overlooked in the AD community. The study is well designed and well described. I have the following comments:
The authors use the term “defective” both in the title and throughout the manuscript. I oppose the use of this terminology, as it conveys the notion of a “broken” device or mechanism. We know very little about the cellular underpinnings of their findings, so the authors should rephrase. I very much support the authors´choice of focusing on slow oscillations, and the use of anesthesia is providing a controlled state, in which these oscillations can be studied. However, the authors need to significantly expand the characterization of anesthetic depths between the conditions, i.e. the AD models and controls. They need to provide quantitative evidence, that the brain state was comparable, i.e. was the occurrence rate (it is technically not a frequency, but an up state event) different between the models and the WT? Line 242, what is a “power imbalance”? The authors need to be more precise in describing their findings. Line 284, what is an “instantaneous amplitude”? The point of the nesting of faster oscillations within the SO is highly interesting, and the authors should expand their analysis, this is currently not clearly visualized and explained. Line 404, the author suse the term “premature”. Why are these changes in SO dynamics premature? It has been discussed e.g. by Busche and Konnerth, and early hyperexcitable network states, which lead to dysregulations of slow wave coherence, might represent an early, maladaptive network state, prior to direct cytotoxic effect of plaques.
Author Response
Reviewer 3:
We thank the Reviewer for finding our study well designed and well described.
… The authors use the term “defective” both in the title and throughout the manuscript. I oppose the use of this terminology, as it conveys the notion of a “broken” device or mechanism. We know very little about the cellular underpinnings of their findings, so the authors should rephrase…
Yes, we agree with the Reviewer that the term “defective” might be inappropriate at the current stage of knowledge. We substituted “defective” with “dampened” in the title and throughout the manuscript, wherever possible.
… the authors need to significantly expand the characterization of anesthetic depths between the conditions, i.e. the AD models and controls. They need to provide quantitative evidence, that the brain state was comparable, i.e. was the occurrence rate (it is technically not a frequency, but an upstate event) …
As described in Materials and Methods (page 7, line 10 from bottom) a pre-processing step was introduced to ensure that all the animals were stable during the acquisition procedure. This unbiased procedure allows to identify windows with stable heart and respiration rates where analyses were carried out. Average values of bpm and rpm are now shown in Supp. Fig. 1B for WT and B6.152H mice.
As discussed in paragraph 4.6 - Study limitations - we cannot exclude that anesthesia has different effects in the different tg lines. Moreover, differences in UP/Down states are expected, as reported in AD mice by Busche et al (2015). Notwithstanding, we have carried out a preliminary investigation on UP states of WT and B6.152H mice, following the procedure developed by Tsakanikas et al. (Scientific Reports, 2017) and no difference was found between the two groups in the number and duration of UP states (defined as events longer than 500 ms), measured at two depths: sr-lm (1500 mm) (WT: 87.5 ± 5.7 events and 2.22 ± 0.16 s, B6.152H: 88.6 ± 7.3 events and 1.92 ± 0.34 s) and cortex L4/5 (600 mm) (WT: 92.2 ± 2.6 events and 2.05 ± 0.08 s, B6.152H: 87.3 ± 7.3 events and 2.05 ± 0.19 s).
These preliminary data cannot be included in this study for many reasons: i) the software employed was developed by the above mentioned group (I. Skaliora and coll.), that is not involved in this specific project; ii) full analysis of UP/Down states of a large amount of data, such as those obtained with multi-site probe, requires implementation of that software, not available yet, but now under development in collaboration with I. Skaliora; iii) that type of analysis is beyond the aim of the present work.
…Line 242, what is a “power imbalance”?...
We have now better defined “power imbalance”: please see the revised manuscript (page 11, paragraph 3.2, line 4 from bottom) (…With respect to WT mice, the alterations in the relative power distribution represent power imbalances. In particular, as an index of power imbalance…)
…The authors need to be more precise in describing their findings. Line 284, what is an “instantaneous amplitude”?...
We have now introduced a definition: see page … To identify alterations in resting functional connectivity in young, asymptomatic AD mice, we used the cross-correlation of the “instantaneous amplitudes”, i.e., the amplitudes of the complex Hilbert transform of LFPs, as previously described [43]…This procedure allows to convey in a single parameter (instantaneous amplitude) information on the amplitude, phase and frequency of the oscillation.
…The point of the nesting of faster oscillations within the SO is highly interesting, and the authors should expand their analysis, this is currently not clearly visualized and explained…
The Reviewer is right: further analysis should have been done and indeed we expanded our observation to delta waves for comparison with SO. We have now reported that PAC between delta waves in the cortex and higher frequencies in the hippocampus (delta cross-regional PAC) is not changed in the regions where SO PAC is modified in B6.152H mice. This piece of information has now been added (page 18, paragraph 3.4, line 5 from top), see also Fig. 9 Summary. For better presentation, panel E in the original Supp. Fig. 6 (now Supp. Fig. 7), was moved to panel D of Supp. Fig. 8. The Supp. Figure Legends were changed accordingly.
… Line 404, the author uses the term “premature”. Why are these changes in SO dynamics premature? It has been discussed e.g. by Busche and Konnerth, and early hyperexcitable network states, which lead to dysregulations of slow wave coherence, might represent an early, maladaptive network state, prior to direct cytotoxic effect of plaques.
The reviewer is right, we have removed the term “premature” and substituted with “early” to indicate the initial stage of amyloidosis. We have no indication whether the disruption of SO connectivity occurs before the start of hyperexcitability and the causal relationship between the two phenomena. Indeed, at 3 months of age, disruption of SO connectivity occurs in the absence of major effects on the higher frequencies. In contrast, with respect to 3-month-old mice, at 6 months, the connectivity loss did not worsen, while the hyperexcitability increased, especially in theta and beta relative powers, possibly suggesting that SO alterations anticipate hippocampal hyperexcitability. Because these are only correlations, with no causal relationships, we only introduced in the Discussion a more general statement on this issue with reference to the work by Busche and coll. (2015) (see page 21, paragraph 4.1, line 10 from top).

Reviewer 4 Report
The authors of the suggested manuscript present some interesting results considering the network connectivity in established Alzheimer's experimental models. Although this work overall seems important, some things considering the structure of the manuscript should change.
-Inevitably the manuscript contains a lot of technical terms and lot of results. However, it is very complicating for the reader to follow the sequence of the manuscript. Therefore, I think that the results presentation should be simplified. Maybe the authors should consider leaving some details out? or make a summarizing table with all the technical numerical results? (e.g. percentages of alterations)
-I would recommend for the authors to insert a conclusive image/table that summarizes their results per animal model used because now it is a little bit chaotic for a reader that is not so acquainted with the techniques used in the manuscript.
-Abstract should be reformed in order to contain a more conclusive sentence highlighting the importance of the conclusions, because as it is now does not provide the essence of the manuscript.
Author Response
Reviewer 4:
We thank the reviewer for the interest in our manuscript.
…-Inevitably the manuscript contains a lot of technical terms and lot of results. However, it is very complicating for the reader to follow the sequence of the manuscript. Therefore, I think that the results presentation should be simplified. Maybe the authors should consider leaving some details out? or make a summarizing table with all the technical numerical results? (e.g. percentages of alterations) …
The manuscript is indeed dense with results and a lot of technical terms. Because of the type of data source (multisite recording probe) we had to manage a significant amount of data, both in time and space. We tried to reduce these data by selecting only 7 out of 24 channels, without losing too much information. We also tried to maintain a balance between simplicity and precision: we thus preferred to introduce quantitative data in terms of percentages along the text, while the absolute values were shown only in Supp. Figures 3 and 4.
In this revised version, we did not substantially modify the Results session since the other Reviewers did not raise this issue, but we agree that introducing a summarizing table is extremely helpful for the reader. The new Fig. 9 is an overview of the major changes occurring in the biochemical and electrophysiological parameters of the different tg lines. The summary is only qualitative, just to highlight the progression of the network changes in B6.152H mice, the partial similarity with the other AD models, the PS2.30H and the APPSwe lines, and the distance from the PS2KO line.

Round 2
Reviewer 2 Report
In the revised manuscript, authors have sufficiently addressed the reviewers' concerns.
Reviewer 3 Report
The authors addressed all of my concerns. This is a beautiful paper, I strongly recommend publication.
Reviewer 4 Report
The authors have adressed the majority of my concerns and have improved the presentation and the interpretation of the results.